# ABCB1 and ABCG2 Control Brain Accumulation and Intestinal Disposition of the Novel ROS1/TRK/ALK Inhibitor Repotrectinib, While OATP1A/1B, ABCG2, and CYP3A Limit Its Oral Availability

**DOI:** 10.3390/pharmaceutics13111761

**Published:** 2021-10-21

**Authors:** Wenlong Li, Rolf W. Sparidans, Maria C. Lebre, Jos H. Beijnen, Alfred H. Schinkel

**Affiliations:** 1The Netherlands Cancer Institute, Division of Pharmacology, Plesmanlaan 121, 1066 CX Amsterdam, The Netherlands; w.li@nki.nl (W.L.); c.lebre@nki.nl (M.C.L.); j.beijnen@nki.nl (J.H.B.); 2Division of Pharmacology, Department of Pharmaceutical Sciences, Faculty of Science, Utrecht University, Universiteitsweg 99, 3584 CG Utrecht, The Netherlands; R.W.Sparidans@uu.nl; 3Division of Pharmacoepidemiology & Clinical Pharmacology, Department of Pharmaceutical Sciences, Faculty of Science, Utrecht University, Universiteitsweg 99, 3584 CG Utrecht, The Netherlands; 4The Netherlands Cancer Institute, Department of Pharmacy & Pharmacology, Plesmanlaan 121, 1066 CX Amsterdam, The Netherlands; 5Alfred H. Schinkel, Schinkel Group, Division of Pharmacology, The Netherlands Cancer Institute, Plesmanlaan 121, 1066 CX Amsterdam, The Netherlands

**Keywords:** repotrectinib (TPX-0005), P-glycoprotein, breast cancer resistance protein, brain accumulation, organic anion transporting polypeptides, cytochrome P450-3A, oral availability

## Abstract

Repotrectinib shows high activity against ROS1/TRK/ALK fusion-positive cancers in preclinical studies. We explored the roles of multidrug efflux transporters ABCB1 and ABCG2, the OATP1A/1B uptake transporter(s), and the CYP3A complex in pharmacokinetics and tissue distribution of repotrectinib in genetically modified mouse models. In vitro, human ABCB1 and ABCG2, and mouse Abcg2 efficiently transported repotrectinib with efflux transport ratios of 13.5, 5.6, and 40, respectively. Oral repotrectinib (10 mg/kg) showed higher plasma exposures in Abcg2-deficient mouse strains. Brain-to-plasma ratios were increased in *Abcb1a/1b^−/−^* (4.1-fold) and *Abcb1a/1b;Abcg2^−/−^* (14.2-fold) compared to wild-type mice, but not in single *Abcg2^−/−^* mice. Small intestinal content recovery of repotrectinib was decreased 4.9-fold in *Abcb1a/1b^−/−^* and 13.6-fold in *Abcb1a/1b;Abcg2^−/−^* mice. Intriguingly, *Abcb1a/1b;Abcg2^−/−^* mice displayed transient, mild, likely CNS-localized toxicity. Oatp1a/1b deficiency caused a 2.3-fold increased oral availability and corresponding decrease in liver distribution of repotrectinib. In *Cyp3a^−/−^* mice, repotrectinib plasma AUC_0–h_ was 2.3-fold increased, and subsequently reduced 2.0-fold in humanized CYP3A4 transgenic mice. Collectively, Abcb1 and Abcg2 restrict repotrectinib brain accumulation and possibly toxicity, and control its intestinal disposition. Abcg2 also limits repotrectinib oral availability. Oatp1a/1b mediates repotrectinib liver uptake, thus reducing its systemic exposure. Systemic exposure of repotrectinib is also substantially limited by CYP3A activity. These insights may be useful to optimize the therapeutic application of repotrectinib.

## 1. Introduction

Chromosomal rearrangements involving ROS proto-oncogene 1 (ROS1) and the anaplastic lymphoma kinase (ALK) gene have been associated with many types of malignancies, including non-small cell lung cancer (NSCLC) [1,2]. There is a high level of amino acid sequence homology between ROS1 and ALK, and their rearrangements represent promising molecular targets for treatment. Several ROS1- and/or ALK-targeting agents have been developed [3,4]. Crizotinib (Xalkori) was the first ALK/ROS1 inhibitor approved by the FDA for metastatic NSCLC, in 2011. More recently, gene fusions involving neurotrophic tropomyosin receptor kinases (NTRKs) have also been shown to be oncogenic drivers in various types of pediatric and adult cancer [5,6,7]. Larotrectinib (Vitrakvi) and entrectinib (Rozlytrek) have been approved for the treatment of solid tumors with TRK rearrangement independent of age and tumor histology. Although these earlier-generation tyrosine kinase inhibitors (TKIs) demonstrated markedly improved clinical outcomes, relapse inevitably develops, and subsequent therapeutic options overcoming acquired resistance remain limited [8,9].

Repotrectinib (TPX-0005; Figure 1A), a next-generation ROS1/TRK/ALK TKI, was specifically designed to combat clinical resistance mutations in the ROS1, TRK and ALK kinases, especially the solvent-front and gatekeeper mutations, by efficiently binding to the active kinase conformation and avoiding steric interference [10]. In line with this, highly potent antiproliferative activity of repotrectinib against wild-type and solvent-front mutated ROS1, TRK, and ALK has been observed in cellular inhibition assays and xenograft models in multiple preclinical models [10,11,12]. Moreover, preliminary TRIDENT-1 data support repotrectinib as a potential best-in-class treatment in ROS1-positive advanced NSCLC, with an 86% confirmed objective response rate (ORR) in TKI-naïve ROS1 + NSCLC and 50% in TKI-pretreated patients [13]. Furthermore, prominent clinical activity of repotrectinib was also seen in 50% of patients who had disease progression on prior TRK inhibitors [13,14]. The majority of the treatment-related adverse events were manageable and limited to grade 1 or grade 2. Repotrectinib for pediatric and adult patients with advanced solid tumors harboring ROS1, TRK, and ALK rearrangements is currently in phase I/II clinical trials (NCT03093116; NCT04094610) [12].

Transmembrane transporters can act as important determinants of the pharmacokinetics, and hence the efficacy and safety profile of many drugs. Multispecific drug transporters belong to two main superfamilies: ATP-binding cassette (ABC) transporters and solute carrier (SLC) transporters [15]. The ABC drug efflux transporters P-glycoprotein (ABCB1; MDR1) and breast cancer resistance protein (ABCG2; BCRP), as well as SLC drug uptake transporter organic anion transporting polypeptides 1A/1B (OATP1A/1B) are of profound interest due to their broad substrate specificity and functional expression in pharmacokinetically important tissues, including critical barrier sites (such as blood-brain barrier (BBB)) and excretory organs (liver, kidney, intestine) [16,17,18]. These transporters have also been detected in tumor cells, potentially causing multidrug resistance against anticancer drugs [19,20]. In addition, efflux capacity mediated by ABCB1 and ABCG2 in brain endothelial capillary cells of the BBB helps to protect the central nervous system (CNS) from harmful compounds, but it also limits the CNS exposure of anticancer agents [21,22]. This might weaken their efficacy against brain tumors (e.g., brain metastasis and gliomas) that often also include ROS1, ALK, and NTRK fusion-positive cancers. The possible transport of repotrectinib by ABCB1, ABCG2, and OATP1A/1B transporters could be of concern because this might lead to interindividual variability in pharmacokinetics. This could affect therapeutic efficacy, due to genetic polymorphisms or transporter-mediated drug–drug interactions, or contribute to adverse drug reactions.

Drug transporters often modulate drug absorption and availability, distribution, and elimination together with drug-metabolizing enzymes. Cytochrome P450 (CYP) is a major metabolizing enzyme superfamily, among which the CYP3A enzymes account for the metabolism of around 50% of currently used drugs [23]. The plasma exposure and oral availability of their substrates can be markedly affected by CYP3A activity, which thus may substantially influence the therapeutic efficacy and even toxicity in patients. Given the high intra- and interindividual variation in CYP3A activity, in patients it is of importance to investigate the extent of interaction between CYP3A and repotrectinib. This is currently unknown based on publicly available sources.

Here, we studied the roles of the multidrug transporters ABCB1, ABCG2, and OATP1A/1B in repotrectinib plasma exposure and tissue accumulation, using wild-type and genetically modified knockout and transgenic mouse models. Moreover, we evaluated to what extent CYP3A affects the oral availability of repotrectinib.

## 2. Materials and Methods

### 2.1. Materials

Repotrectinib (TPX-0005; 100%) was purchased from TargetMol (Wellesley Hills, MA, USA). Ko143 was supplied by Tocris Bioscience (Bristol, UK). Zosuquidar was obtained from Sequoia Research Products (Pangbourne, UK). Bovine serum albumin (BSA) fraction V was supplied by Roche Diagnostics (Mannheim, Germany). Heparin (5000 IU/mL) and isoflurane were obtained from Leo Pharma (Breda, The Netherlands) and Pharmachemie (Haarlem, The Netherlands), respectively. All other chemicals and reagents were purchased from Sigma-Aldrich (Steinheim, Germany). Chemicals that were used in the bioanalytical repotrectinib method were previously reported [24].

### 2.2. Cell Lines and Transport Assays

Polarized Madin-Darby Canine Kidney (MDCK-II) cells (ECACC 00062107) stably transduced with either human (h) ABCB1, hABCG2 or mouse (m) Abcg2 cDNA were generated in our institute between 1995–2005. The proper identity and functionality, such as highly characteristic growth and drug transporter properties, including inhibitor sensitivity, were regularly checked in these epithelial cells. Mycoplasma routinely tested negative in these cells. The passage number was 10–15 when used in the transport experiments.

Transepithelial transport experiments were performed as previously described with 12-well plates (Transwell 3414, Corning Inc., Kennebunk, ME, USA) containing microporous polycarbonate membrane filters (3.0 µm pore size, 12 mm diameter). The parental MDCK-II cells and their subclones were seeded on the membrane at a density of 2.5 × 10^5^ cells in each well and maintained and grown in DMEM with 10% fetal bovine serum (FBS, Sigma), 100 U/mL penicillin, and 100 µg/mL streptomycin, at 37 °C in 5% CO_2_ and 95% humidified air, for three days to form an intact monolayer. Prior to and after the transport phase, the integrity of monolayer membrane was checked and confirmed by the measurement of the transepithelial electrical resistance (TEER).

After pre-incubation with 5 µM zosuquidar (ABCB1 inhibitor) and/or 5 µM Ko143 (ABCG2/Abcg2 inhibitor) in both compartments (when appropriate) for 1 h, the transport phase was initiated (t = 0) by replacing the medium in apical and basolateral compartments with fresh DMEM with 10% FBS and the appropriate inhibitor(s), and repotrectinib at 5 µM in the donor compartment. The cells were kept at 37 °C, pH~7.4, in a 5% CO_2_ and 95% humidified environment during the experiment. At 1-, 2-, 4-, and 8-h time points, 50 µL aliquots were taken from the acceptor compartment and stored at −30 °C until LC-MS/MS measurement of repotrectinib. The active transport ratio *r*, calculated by dividing the amount of apically directed drug transport by basolaterally directed drug translocation after 8 h, was used to assess active transport of repotrectinib.

### 2.3. Animals

Standard animal housing and care were according to the institutional guidelines complying with Dutch and EU legislations. All experimental animal protocols, including power calculations, designed under the nationally approved DEC/CCD project AVD301002016595, were evaluated and approved by the Institutional Animal Care and Use Committee. Based on the principles of 3Rs (replacement, refinement, and reduction), experimental procedures were optimized. Animal experiments are reported also in compliance with the ARRIVE 2.0 guidelines. In view of the intended clinical application of repotrectinib, either male or female mice were used in different parts of our studies, to minimize the unnecessary culling of mice, and in order to get experimental data for both genders and thus avoid systematic gender bias in our study.

Wild-type, *Abcb1a/1b^−/−^*, *Abcg2^−/−^*, *Abcb1a/1b;Abcg2^−/−^*, *Oatp1a/1b^−/−^*, *Cyp3a^−/−^* and Cyp3aXAV mice [18,25,26], of comparable (> 99% FVB) genetic background, between 9 and 15 weeks of age, with body weights in the range of 25.2–39.7 g, were used. Mice with similar average ages (and body weights) were randomly allocated in experimental groups as far as possible. Mice were maintained in a temperature-controlled and specific pathogen-free environment with a 12-h light and 12-h dark cycle and they received a standard diet (Transbreed, SDS Diets, Technilab-BMI, Someren, The Netherlands) and acidified water ad libitum. Animal welfare was assessed prior to, during, and after the experiments; mice would be humanely sacrificed when they showed discomfort levels higher than mild. Given the objective main readout (repotrectinib plasma and tissue concentrations as measured by LC-MS/MS), no blinding method was applied.

### 2.4. Drug Stock and Working Solution

Repotrectinib was dissolved in dimethyl sulfoxide (DMSO) at a concentration of 50 mg/mL. This stock solution was first diluted 3-fold with ethanol/polysorbate 80 (1/1; *v*/*v*), followed by further dilution with a 10 mM hydrochloric acid solution, yielding a dosing solution of 1 mg/mL, freshly prepared on the day of experiment. Final concentrations of the solvents in the administration solution were: 2%, 2%, 2%, and 94% (all *v*/*v*) for DMSO, ethanol, polysorbate 80, and 10 mM hydrochloric acid solution, respectively.

### 2.5. Plasma Pharmacokinetics and Tissue Distribution of Repotrectinib in Mice

To minimize variation in absorption upon oral administration, mice were fasted for 2–3 h before repotrectinib (10 mg/kg body weight) was administered by gavage into the stomach at 10 µL/g body weight, with a blunt-ended needle. Tail vein serial blood samples were collected at 0.125, 0.25, 0.5, 1, 2 and 4 h using heparinized capillary tubes (Sarstedt, Nümbrecht, Germany) for the 8-h experiments. For the 4- and 2-h experiments, blood samplings were performed from the tail vein with the same schedule, but terminated at 2 and 1 h, respectively. Eight, four, or two hours after oral administration, mice were deeply anesthetized with 2–3% isoflurane, followed by cardiac puncture to collect blood in Eppendorf tubes containing heparin as an anticoagulant. Brain, liver, spleen, kidney, lung, small intestine, and testis were rapidly collected after sacrificing the mice by cervical dislocation. The small intestinal contents (SIC) were separated from small intestinal tissue (SI), which was flushed and rinsed with cold saline to remove any residual feces. Brain, liver, spleen, kidney, lung, SI, SIC, and testis were homogenized with 1, 3, 1, 2, 1, 3, 2, and 1 mL of 2% BSA (*w*/*v*), respectively. Plasma fraction was isolated from blood by centrifugation at 9000× *g* for 6 min at 4 °C. All samples were stored at −30 °C until analysis.

### 2.6. LC-MS/MS Analysis

A sensitive and specific liquid chromatography-tandem mass spectrometry method using protein precipitation pre-treatment was adopted to quantify the concentration of repotrectinib in DMEM medium, plasma samples and tissue homogenates [24].

### 2.7. Data and Statistical Analysis

A non-compartmental model using the PKSolver add-in program for Microsoft Excel was adopted to calculate the pharmacokinetic parameters of repotrectinib [27]. Oral availability was assessed by the plasma area under the curve (AUC), calculated using repotrectinib plasma concentration-time curves with the linear trapezoidal rule without extrapolating the infinity. Statistical testing was performed in Graphpad Prism7 (Graphpad Software, La Jolla, CA, USA). All data are presented as geometric means ± SD. Heteroscedastic data were log-transformed before applying statistical analysis. When differences between two groups were compared, the two-sided unpaired Student’s *t* test was used. One-way analysis of variance (ANOVA) was applied when multiple groups were compared, and the Bonferroni post hoc correction was used to accommodate multiple testing. Differences were considered statistically significant when *P* < 0.05.

## 3. Results

### 3.1. Repotrectinib Is Efficiently Transported by Human ABCB1 and ABCG2, and by Mouse Abcg2 In Vitro

Transepithelial drug transport was evaluated using polarized monolayers of MDCK-II parental cells and its subclones stably overexpressing hABCB1, hABCG2 or mAbcg2. Repotrectinib at 5 µM was modestly transported in the apical direction (efflux transport ratio *r* = 2.5, Figure 2A) in the parental cells, and addition of the ABCB1 inhibitor zosuquidar completely abrogated this transport (*r* = 1.0, Figure 2B). These data indicate that the low-level endogenous canine ABCB1 present in MDCK-II cells could modestly transport repotrectinib. In the hABCB1-overexpressing cells, the pronounced apically directed transport of repotrectinib (*r* = 13.5, Figure 2C) was extensively reduced by adding zosuquidar (*r* = 1.1, Figure 2D). In the subsequent experiments with hABCG2- and mAbcg2-overexpressing cells, zosuquidar was applied to circumvent any transport contribution from canine ABCB1. Repotrectinib was highly transported by cells overexpressing mAbcg2 and hABCG2, with efflux transport ratios of 40 and 5.6 (Figure 2G,E), respectively. The transport activity of mAbcg2 and hABCG2 was fully inhibited by the addition of Ko143, a specific ABCG2 inhibitor (Figure 2H,F). Repotrectinib thus appears to be very efficiently transported by human ABCB1 and ABCG2, and mouse Abcg2, and modestly by canine ABCB1.

### 3.2. Both ABCB1 and ABCG2 Control Repotrectinib Plasma Exposure and Tissue Distribution

To study the possible roles of ABCB1 and ABCG2 in controlling plasma exposure and tissue accumulation of repotrectinib, an 8-h pilot experiment was performed in female wild-type and *Abcb1a/1b;Abcg2^−/−^* mice, at a dose of 10 mg/kg repotrectinib by oral administration. As shown in Appendix A, repotrectinib was rapidly absorbed, with peak plasma concentrations seen at 30 min after dosing. *Abcb1a/1b;Abcg2^−/−^* mice showed a significant 1.4-fold increase in plasma exposure of repotrectinib over 8 h (AUC_0–8h_) relative to wild-type mice.

Moreover, we observed a very substantial increase in brain concentrations (19.5-fold) and brain-to-plasma ratios (9.5-fold) of repotrectinib in *Abcb1a/1b;Abcg2^−/−^* mice compared to wild-type mice (Appendix A). Interestingly, the total amount of repotrectinib in small intestinal content (SIC) was markedly reduced in *Abcb1a/1b;Abcg2^−/−^* mice relative to wild-type mice (Appendix A). This suggests that there was likely a more extensive absorption of repotrectinib when ABC transporters were absent. This could result from a reduced enterohepatic recirculation of absorbed repotrectinib through hepatobiliary excretion in the bile canaliculi of liver, or from increased net absorption across the intestinal wall (essentially due to loss of an intestinal excretion function), or a combination of both processes. The SI tissue distribution of repotrectinib between the mouse strains probably simply followed the profiles seen in SIC, albeit with around 10-fold lower concentration levels (Appendix A). For the other tested tissues, including liver, spleen, kidney, and lung, no significant differences in tissue-to-plasma ratios were found between wild-type and *Abcb1a/1b;Abcg2^−/−^* mice (Appendix A).

Of note, during the experiment we observed mild toxicity signs in *Abcb1a/1b;Abcg2^−/−^* mice, including hypoactivity and continuously closed eyes, at 1–2 h after oral administration. These signs of toxicity disappeared around 4 h. In contrast, no noticeable toxicity symptoms were found in wild-type mice.

To further investigate the separate or combined impact of ABCB1 and ABCG2 on plasma pharmacokinetics and tissue distribution of oral repotrectinib at 10 mg/kg, male wild-type, *Abcb1a/1b^−/−^*, *Abcg2^−/−^*, and *Abcb1a/1b;Abcg2^−/−^* mice were tested. This experiment was terminated at 4 h to achieve relatively high plasma levels of repotrectinib, which are more relevant for assessing the overall impact of ABC transporters on repotrectinib exposure. As shown in Figure 3A,B and Table 1, significantly higher oral availability of repotrectinib was found in *Abcg2^−/−^* and *Abcb1a/1b;Abcg2^−/−^* mice compared to wild-type mice, with significantly higher peak plasma concentrations, but not in single *Abcb1a/1b^−/−^* mice. Moreover, there were no significant differences in plasma AUC_0–4h_ of repotrectinib between *Abcg2^−/−^* and *Abcb1a/1b;Abcg2^−/−^* mice. These data together indicate that mAbcg2 could limit repotrectinib plasma exposure, but mAbcb1a/1b could not, at least not to a significant extent.

Significantly higher brain concentrations of repotrectinib at 4 h after oral administration were observed in single *Abcb1a/1b^−/−^* mice (4.1-fold) and especially *Abcb1a/1b;Abcg2^−/−^* mice (29.1-fold) than in wild-type mice, but not in single *Abcg2^−/−^* mice. Wild-type mice showed a relatively low brain-to-plasma ratio (0.15), which was substantially increased by 4.1-fold in *Abcb1a/1b^−/−^* mice and 14.2-fold in combination *Abcb1a/1b;Abcg2^−/−^* mice (both *P* < 0.001). These results suggest that mAbcb1 could strongly limit the brain distribution of repotrectinib, whereas mAbcg2 also contributes to this process, but the latter effect can only be clearly observed when mAbcb1 is absent. Interestingly, similar signs of transient toxicity as before were observed in *Abcb1a/1b;Abcg2^−/−^* mice, but not in any of the other strains. This suggests a possible role of the dramatically increased brain concentration of repotrectinib in the *Abcb1a/1b;Abcg2^−/−^* strain. Qualitatively similar tissue distribution results were observed for the testis, where Sertoli cells form a similar physical and biochemical protective barrier as the brain, the blood-testis barrier (BTB). The testis-to-plasma ratios of repotrectinib were increased by 5.6-fold when both ABC transporter systems were absent, suggesting that the relative impact of mAbcb1a/1b and mAbcg2 was somewhat lower than that in the brain (Appendix A and Table 1).

Furthermore, modestly but significantly lower liver-to-plasma ratios of repotrectinib were found in *Abcg2^−/−^* and *Abcb1a/1b;Abcg2^−/−^* mice compared to wild-type mice, but not in single *Abcb1a/1b^−/−^* mice (Figure 3F and Table 1). This might suggest a reduced concentration of repotrectinib in the intrahepatic bile of Abcg2-deficient mice. On the other hand, the amount of repotrectinib in SIC of wild-type mice represented about 5.6% of the total oral dose, and was markedly and significantly reduced in both *Abcb1a/1b^−/−^* (1.1%) and *Abcb1a/1b;Abcg2^−/−^* mice (0.4%) compared to the other strains (Figure 3G and Table 1). Moreover, compared to single *Abcg2^−/−^* mice (4.8%), the recovery of repotrectinib was significantly lower in *Abcb1a/1b;Abcg2^−/−^* mice. These data suggest that Abcg2 may affect primarily biliary excretion, and Abcb1a/1b primarily intestinal efflux of repotrectinib. A similar profile of repotrectinib distribution was seen in the SI tissue of these strains, whereas in spleen and kidney no significant differences were observed when assessing tissue-to-plasma ratios (Appendix A).

### 3.3. Mouse Oatp1a/1b Controls Oral Availability, Liver, and Small Intestine Distribution of Repotrectinib

OATP-mediated uptake can impact the oral availability and tissue distribution of drugs. The possible interactions of repotrectinib with OATP/SLCO uptake transporters are largely unknown. To study this aspect, a pilot experiment administering repotrectinib (10 mg/kg) orally to female wild-type and *Oatp1a/1b^−/−^* mice was performed up to 8 h. The plasma and tissue levels of repotrectinib were analyzed. As shown in Appendix A, the systemic exposure over 8 h (AUC_0–8h_) was substantially increased by 1.5-fold (*P* < 0.05) in *Oatp1a/1b^−/−^* mice compared to wild-type mice. However, liver-to-plasma ratios were not significantly different between wild-type and *Oatp1a/1b^−/−^* mice at 8 h, when the plasma levels of repotrectinib in both mouse strains were much reduced compared to the peak concentration.

To obtain a more complete plasma and tissue concentration-time profile at higher plasma levels, a follow-up experiment using male wild-type and *Oatp1a/1b^−/−^* mice was terminated at 4 h after oral administration of 10 mg/kg repotrectinib. In *Oatp1a/1b^−/−^* mice, the plasma AUC_0–4h_ of repotrectinib was significantly increased by 2-fold compared to that in wild-type mice (Appendix A). In line with these results, the liver-to-plasma ratios were markedly decreased (2.5-fold) in *Oatp1a/1b^−/−^* mice (Appendix A). Moreover, in *Oatp1a/1b^−/−^* mice, the percentage of dose of repotrectinib recovered in the SIC corrected for plasma concentration was slightly but significantly reduced in Oatp1a/1b-deficient mice (Appendix A). A similar profile was observed in the SI tissue distribution (Appendix A). In contrast, no significant differences were observed in the relative tissue distributions of repotrectinib in other tested tissues (brain, testis, kidney, spleen) between the two strains (Appendix A).

To further study the relevance of these phenomena at a higher plasma concentration, a 2-h experiment was performed in male wild-type and *Oatp1a/1b^−/−^* mice. Assuming that the absorption phase of repotrectinib in small intestine was most likely completed 2 h after oral administration, this allowed us to study the impact of Oatp1a/1b on repotrectinib enterohepatic circulation. The small intestine with content (SIWC) combined was collected given the similarity in results between these two compartments obtained in previous experiments. As shown in Figure 4 and Table 2, in the absence of Oatp1a/1b oral availability and liver-to-plasma ratios of repotrectinib were substantially increased and reduced (in each case by 2.3-fold, *P* < 0.001), respectively, compared to these parameters in wild-type mice. Importantly, while a similar amount of repotrectinib (% of dose) was recovered in the SIWC, after correction for plasma concentration or plasma exposure (AUC_0–2h_), the relative intestinal (SIWC) recovery of repotrectinib was significantly decreased by 2.8-fold and 1.9-fold, respectively, in *Oatp1a/1b^−/−^* mice compared to that in wild-type mice. Overall, these data demonstrate that mouse Oatp1a/1b transporters could mediate substantial uptake of repotrectinib into the liver, and thus markedly control the oral availability, liver distribution, and enterohepatic circulation of repotrectinib.

### 3.4. CYP3A Markedly Limits Repotrectinib Systemic Exposure

To clarify the possible interactions between CYP3A and repotrectinib, we performed an 8-h experiment in female wild-type, *Cyp3a^−/−^* (Cyp3a knockout) and Cyp3aXAV mice, which are specifically overexpressing human CYP3A4 in the liver and intestine of *Cyp3a^−/−^* mice. Serial blood samples at different time points and organs at 8 h were collected and processed as described above. The plasma exposure of repotrectinib was significantly increased by 2.3-fold in *Cyp3a^−/−^* mice relative to wild-type mice. In addition, the repotrectinib plasma AUC_0–8h_ in Cyp3aXAV mice was markedly decreased by 2.0-fold compared to *Cyp3a^−/−^* mice (Figure 5 and Table 3). Interestingly, the elimination half-life was significantly reduced in *Cyp3a^−/−^* mice, but then restored to wild-type levels in the Cyp3aXAV mice (Figure 5B). These data suggest that both mouse Cyp3a and human CYP3A4 play an important role in hepatic metabolism of repotrectinib, and thus could limit the systemic exposure of repotrectinib, to roughly similar extents.

In contrast to the substantial effects on plasma exposure, the tissue-to-plasma ratios of repotrectinib were not significantly altered between wild-type, *Cyp3a^−/−^* and Cyp3aXAV mice for brain, SIC, SI, liver, spleen, kidney, and lung, indicating little or no impact of CYP3A activity on the relative tissue distribution (Appendix A and Table 3). Therefore, these results indicate that repotrectinib is markedly metabolized by mouse Cyp3a and human CYP3A4, which profoundly affects its oral availability, but not its relative tissue distribution.

## 4. Discussion

Repotrectinib was found to be efficiently transported in vitro by human ABCB1, human ABCG2, and mouse Abcg2, and modestly by canine ABCB1. The oral availability of repotrectinib was significantly increased in *Abcg2^−/−^* and *Abcb1a/1b;Abcg2^−/−^* mice, but not in single *Abcb1a/1b^−/−^* mice, compared to wild-type mice. Moreover, high increases in brain-to-plasma ratios were observed in the *Abcb1a/1b^−/−^* (4.1-fold) and *Abcb1a/1b;Abcg2^−/−^* (14-fold) mice. These transporters in the BBB thus actively keep repotrectinib out of the brain, with ABCB1 playing a dominant role, and may also restrict its CNS toxicity. A similar impact was observed for repotrectinib distribution in the testis. While somewhat indirect, liver concentration and SIC data suggested that Abcg2 plays a substantial role in biliary excretion/concentration, and Abcb1a/1b in direct intestinal efflux of repotrectinib. Markedly increased systemic exposure and decreased liver distribution of repotrectinib in *Oatp1a/1b^−/−^* compared to wild-type mice indicated that mouse Oatp1a/1b transporters can mediate the hepatic uptake of repotrectinib. Repotrectinib oral availability, but not relative tissue distribution, was further restricted by mouse Cyp3a and human CYP3A4. This shows that CYP3A is an important factor in determining metabolic clearance of repotrectinib in both species.

After oral administration of repotrectinib, the absorption among all tested mouse strains was relatively fast, with a T_max_ around 30 min, followed by gradual elimination. Importantly, the systemic exposure in wild-type mice dosed with repotrectinib at 10 mg/kg reached clinically relevant exposure levels, similar to those in patients taking repotrectinib at 160 mg once daily [10]. With the same oral dose of repotrectinib, we observed about 1.6-fold higher plasma levels of repotrectinib in female than in male mice (Figure 3 and Appendix A). This could mean that the expression level and/or efficiency of one or more detoxification systems for repotrectinib is gender-dependent. Of note, however, the relative impact of ABC efflux transporters and OATP1A/1B uptake transporters on repotrectinib pharmacokinetics was not much affected by the gender.

In line with efficient in vitro transport by Abcg2, we observed a significantly increased plasma AUC (1.7 to 2.0-fold) of repotrectinib in Abcg2-deficient strains. Moreover, the liver-to-plasma ratios of repotrectinib were significantly reduced by 30–35% in *Abcg2^−/−^* and *Abcb1a/1b;Abcg2^−/−^* mice, but not in single *Abcb1a/1b^−/−^* mice, compared to wild-type mice. These data may reflect a partially reduced Abcg2-mediated excretion from hepatocytes into bile, thus leading to increased plasma exposure. The relatively reduced liver-to-plasma ratio of repotrectinib in Abcg2 knockout mice may reflect strongly decreased intrahepatic bile accumulation of repotrectinib due to the Abcg2 deficiency. The markedly reduced recovery of repotrectinib in the small intestinal content in the absence of especially Abcb1a/1b and, to a lesser extent, Abcg2 (Figure 3) is likely caused by substantially reduced direct intestinal excretion due to the Abcb1a/1b knockout, and reduced biliary excretion due to the Abcg2 knockout, while there may be some overlap between these two processes for each transporter. In *Abcb1a/1b;Abcg2^−/−^* mice, this combination of increased net intestinal absorption and decreased hepatobiliary excretion of repotrectinib explains the increased plasma AUC and strongly reduced intestinal content levels of repotrectinib (Figure 3).

Considering the therapeutic importance of readily reaching the brain tumors and brain metastases commonly occurring in NSCLC, the interactions between repotrectinib and the ABC transporters in the BBB should be thoroughly investigated. Similar to brigatinib and many other TKIs [22,28], repotrectinib brain penetration in vivo is subject to collaborative action of Abcb1a/1b and Abcg2 at the BBB. The phenomenon that single knockout of one of the transporters (in this case Abcg2) has no detectable effect on brain distribution of a drug, whereas in combination with the other transporter knockout it does have a clear effect, has previously been reported and explained using straightforward pharmacokinetic models [29]. In the case of repotrectinib, Abcb1a/1b clearly has a more pronounced impact than Abcg2 in restricting its brain penetration. Yun et al. (2020) demonstrated that repotrectinib exhibited potent antitumor activity in the CNS with efficient BBB penetrating properties in ROS1+ patient-derived xenograft (PDX) mouse models with brain metastases of NSCLC [12]. While we observed intrinsically modest penetration of repotrectinib into the brain in wild-type mice, with a brain-to-plasma ratio of 0.15, this penetration could be substantially boosted by 14-fold through removal of both Abcb1a/1b and Abcg2 in the BBB. This is comparable to crizotinib, the first ALK/ROS1 inhibitor, that demonstrated a brain-to-plasma ratio of about 0.2 in wild-type mice, which could be enhanced ~15-fold by inhibiting or removing P-glycoprotein [28]. The low brain-to-plasma ratio of crizotinib has led to suggestions that the development of brain metastases frequently observed during treatment with crizotinib may be related to the poor BBB penetration and brain exposure of this drug [30]. By analogy, this suggests that brain metastases may also develop in patients treated with repotrectinib.

CNS metastases represent a major cause of morbidity and mortality in patients with ROS1- and ALK- driven NSCLC. Given the limited brain exposure of repotrectinib found in wild-type mice because of Abcb1a/1b and Abcg2 functions, and their potential clinical relevance for limiting therapeutic efficacy against brain malignancies as well as tumor cells that themselves express ABC transporters, we could consider to use this insight to enhance the CNS (and tumor) exposure of repotrectinib by the application of potent pharmacological ABCB1 and ABCG2 inhibitors such as elacridar.

The mild toxicity of repotrectinib observed in mice in which both Abcb1a/1b and Abcg2 were absent was probably related to on-target adverse events due to TRK inhibition, most likely in the brain (see below). The transient toxicity symptoms were very similar to those recently reported in an on-going phase I clinical trial (NCT03063116), in which dizziness is the most common adverse effects of repotrectinib. This side effect could tentatively have resulted from decreased proprioception and cerebellar dysfunction [31,32].

The toxicity of repotrectinib in mice appears not to be directly related to the plasma concentration and exposure of repotrectinib, as the systemic exposures in *Cyp3a^−/−^* and *Oatp1a/1b^−/−^* mice were at least as high as those in *Abcb1a/1b;Abcg2^−/−^* mice, and yet no noticeable signs of toxicity were observed in either of these strains. The pronounced difference in susceptibility suggests that the markedly increased CNS distribution of repotrectinib is most likely behind this toxicity. Given this apparently mild and manageable toxicity together with the potential benefits for treatment of brain metastases, attempts to improve the CNS distribution of repotrectinib in patients by using an efficacious ABCB1/ABCG2 inhibitor may still be worth considering, but should obviously be carefully monitored for possibly increased toxicity symptoms.

Among ROS1, TRK, and ALK inhibitors approved by the FDA or currently under clinical investigation, there are two other novel macrocyclic TKIs, lorlatinib (ALK/ROS1 inhibitor) and selitrectinib (TRK inhibitor; Figure 1B,C). We showed previously that the brain-to-plasma ratios of lorlatinib and selitrectinib were 0.4 and 0.03 in wild-type mice, respectively, and could be increased by 4- and 6-fold by combined genetic knockout of Abcb1 and Abcg2 [33,34]. Our data suggest that Abcb1 and Abcg2 have a relatively stronger impact on the brain accumulation of repotrectinib, even in spite of its smaller molecule weight than lorlatinib and selitrectinib. This further emphasizes that it remains challenging to predict the intrinsic CNS permeability of compounds just based on their physicochemical properties.

Since repotrectinib is a macrocyclic compound and highly hydrophobic, it is thought to easily pass through cell membranes by passive diffusion. Theoretically, it is less likely that uptake transporters, such as Oatp1a/1b, would be an important factor determining the in vivo pharmacokinetics of repotrectinib. Yet, the present study shows that mouse Oatp1a/1b transporters could substantially control plasma exposure, liver distribution and intestinal disposition of repotrectinib. Absence of Oatp1a/1b activity resulted in a marked increase in repotrectinib oral availability and a corresponding decrease in liver accumulation. Intriguingly, the relative distribution of repotrectinib in small intestine with content was also markedly reduced. The lower small intestinal levels of repotrectinib in Oatp1a/1b-deficent mice may be due to decreased hepatobiliary excretion of repotrectinib as a secondary consequence of the reduced hepatic uptake levels. The observed strong in vivo transport of repotrectinib by Oatp1a/1b may have potentially important applications for the clinic. There is a very likely possibility that patients with variant activities of OATP1A/1B transporters might show different pharmacokinetics, efficacy and even safety profiles of repotrectinib treatment due to the altered exposure of repotrectinib in systemic circulation and/or organs (e.g., less hepatic uptake and thus reduced metabolism and clearance).

Based on publicly available information, little is known about the interaction of repotrectinib with CYP3A. Our experiments clearly demonstrate that both mouse and human CYP3A can markedly reduce the plasma levels and thus overall systemic exposure of repotrectinib. This suggests that variable activity of CYP3A in patients, due to either drug–drug interactions or genetic polymorphisms, would likely have a noticeable impact on metabolic clearance and oral availability of repotrectinib. This probably should be taken into consideration in clinical dosing of repotrectinib for its future broader clinical use. It is worth mentioning that the relative tissue distribution of repotrectinib was similar among wild-type, *Cyp3a^−/−^* and Cyp3aXAV mice. This suggests that coadministration of a CYP3A inhibitor, such as ritonavir, may present an approach to boost the systemic exposure of repotrectinib in patients without invoking the risk of altering relative tissue distribution [35]. In addition, given that the costs of novel targeted anticancer drugs are usually very high, one could even consider to deliberately co-administer ritonavir with an appropriately reduced dose of repotrectinib to achieve therapeutic plasma levels of repotrectinib in patients. For instance, halving the dose of repotrectinib and therefore its cost would already contribute substantial savings in health care costs. However, coadministration of strong CYP3A-inhibiting and perhaps also -inducing drugs with repotrectinib still should be very carefully evaluated and critically monitored in the clinic.

## 5. Conclusions

This is the first study documenting that repotrectinib is transported by ABCB1 and ABCG2 in vitro, that its oral availability is limited by Abcg2, and that Abcb1a/1b and Abcg2 together substantially restrict repotrectinib brain accumulation in mice. Additionally, the absence of Abcb1 and Abcg2 markedly affects the enterohepatic cycling of repotrectinib through the diminished intestinal and/or hepatobiliary excretion of repotrectinib into the lumen of small intestine. Moreover, we demonstrated that, unexpectedly, OATP1A/1B could restrict the plasma exposure of repotrectinib by mediating its hepatic uptake and thus facilitating hepatobiliary excretion. Finally, human CYP3A4 and mouse Cyp3a can markedly reduce the oral availability of repotrectinib, without affecting its relative tissue distribution. These insights may be used to further enhance the therapeutic application, efficacy, and safety of repotrectinib.

## Figures and Tables

**Figure 1 pharmaceutics-13-01761-f001:**
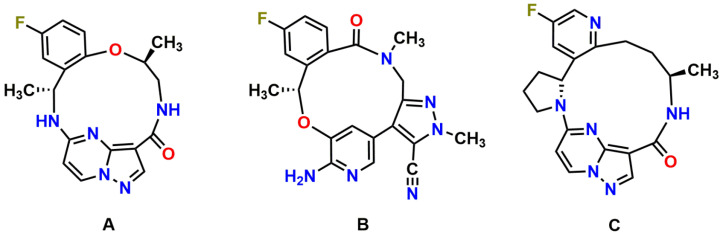
Molecular structures of repotrectinib (**A**) (MW = 355.37 g/mol), lorlatinib (**B**) (MW = 406.42 g/mol), and selitrectinib (**C**) (MW = 380.42 g/mol).

**Figure 2 pharmaceutics-13-01761-f002:**
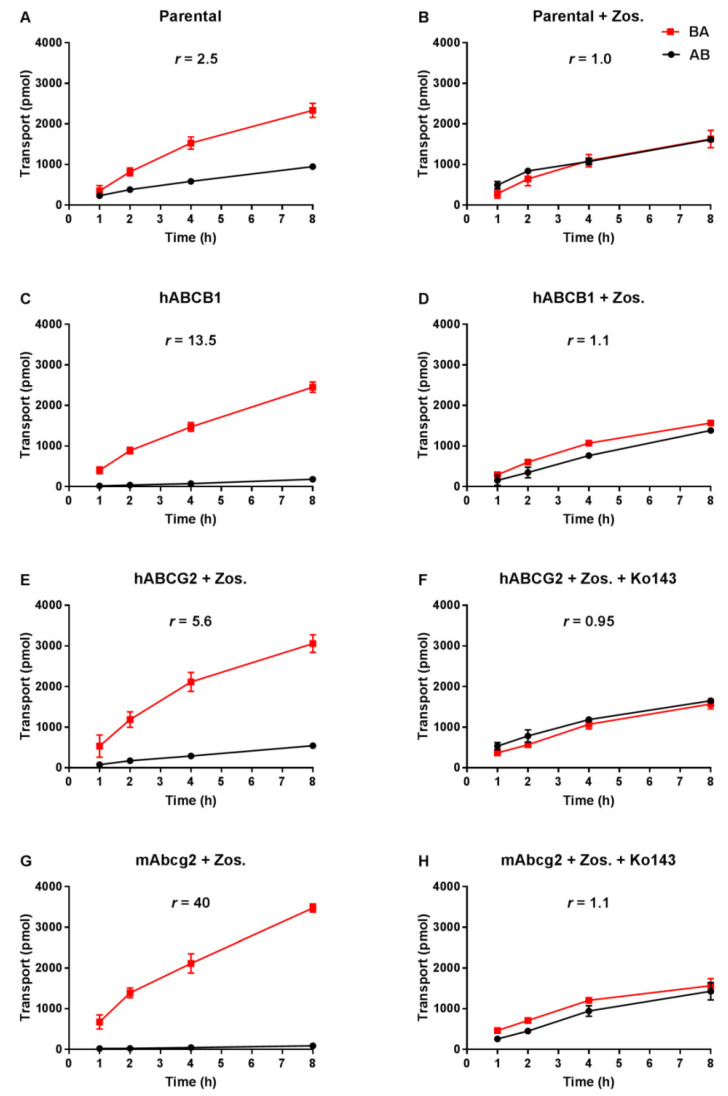
Transepithelial transport of repotrectinib (5 µM) assessed in MDCK-II cells either non-transduced (**A**,**B**), transduced with hABCB1 (**C**,**D**), hABCG2 (**E**,**F**) or mAbcg2 (**G**,**H**) cDNA. At t = 0 h, repotrectinib was applied in the donor compartment (either apical or basolateral) and the concentrations in the acceptor compartment at t = 1, 2, 4, and 8 h were measured and plotted as repotrectinib transport (pmol) in the graph (n = 3). B, D–H: Zos. (zosuquidar, 5 μM) was applied to inhibit human and/or endogenous canine ABCB1. F and H: the ABCG2 inhibitor Ko143 (5 μM) was applied to inhibit ABCG2/Abcg2-mediated transport. *r*, efflux transport ratio. BA (■), translocation from the basolateral to the apical compartment; AB (●), translocation from the apical to the basolateral compartment. Points, mean; bars, S.D. Data are presented as mean ± S.D. (n = 3).

**Figure 3 pharmaceutics-13-01761-f003:**
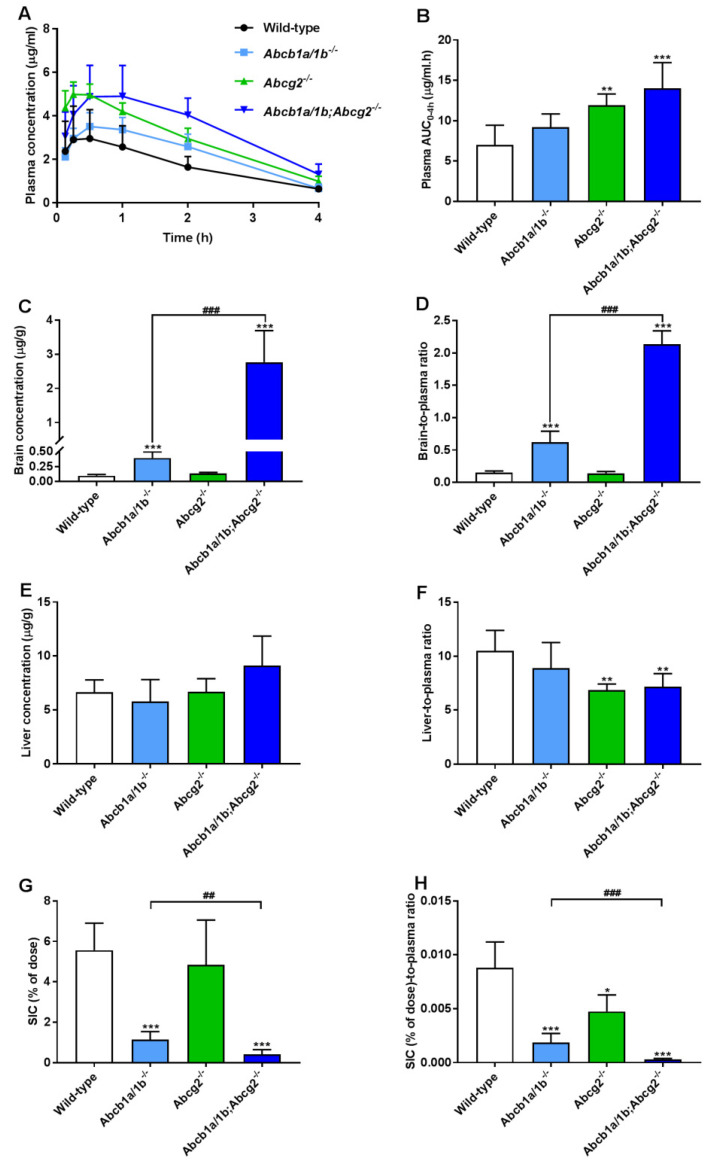
Plasma concentration-time curves (**A**), plasma AUC_0–4h_ (**B**), brain and liver concentration (**C**,**E**), small intestinal content (SIC) as percentage of dose (**G**), and brain-, liver-, SIC (% of dose)-to-plasma ratios (**D**,**F**,**H**) of repotrectinib in male wild-type, *Abcb1a/1b^−/−^*, *Abcg2^−/−^*, and *Abcb1a/1b;Abcg2^−/−^* mice 4 h after oral administration of 10 mg/kg repotrectinib. Due to inhomogeneous variance, brain and SIC data were first log-transformed before applying statistical analysis. SIC (% of dose), drug percentage of dose in small intestinal content expressed as total repotrectinib in SIC divided by total drug administered to the mouse. Data are presented as mean ± S.D. (n = 6–7). *, *P* < 0.05; **, *P* < 0.01; ***, *P* < 0.001 compared to wild-type mice. ^##^, *P* < 0.01; ^###^, *P* < 0.001 comparing *Abcb1a/1b;Abcg2^−/−^* mice to *Abcb1a/1b^−/−^* mice.

**Figure 4 pharmaceutics-13-01761-f004:**
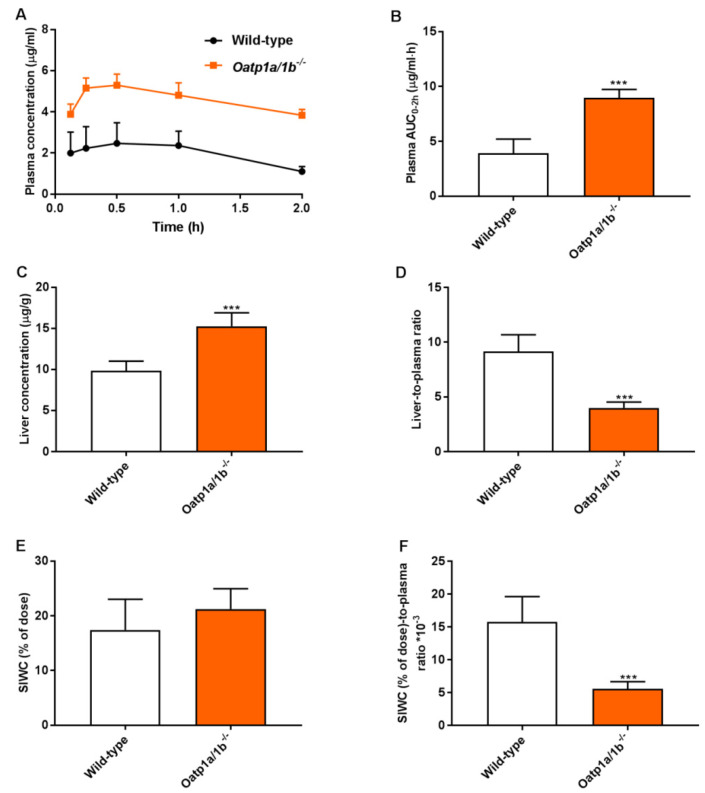
Plasma concentration-time curves (**A**), plasma AUC_0–2h_ (**B**), liver concentration (**C**), small intestine with content (SIWC) as percentage of dose (E), and liver- and SIWC (% of dose)-to-plasma ratios (**D**,**F**) of repotrectinib in male wild-type and *Oatp1a/1b^−/−/−^* mice 2 h after oral administration of 10 mg/kg repotrectinib. SIWC (% of dose), drug percentage of dose in small intestine with content, which was expressed as total repotrectinib in SIWC divided by total drug administered to the mouse. Data are presented as mean ± S.D. (n = 6–7). ***, *P* < 0.001 compared to wild-type mice.

**Figure 5 pharmaceutics-13-01761-f005:**
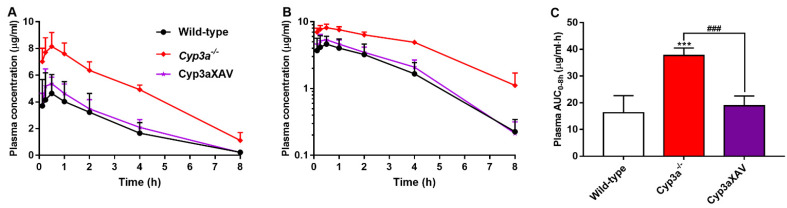
Plasma concentration-time curves (**A**), semi-log plot of plasma concentration-time curves (**B**), and plasma AUC_0–8h_ (**C**) of repotrectinib in female wild-type, *Cyp3a^−/−^*, and Cyp3aXAV mice 8 h after oral administration of 10 mg/kg repotrectinib. Data are presented as mean ± S.D. (n = 6). ***, *P* < 0.001 compared to wild-type mice. ^###^, *P* < 0.001 comparing Cyp3aXAV mice to *Cyp3a^−/−^* mice.

**Table 1 pharmaceutics-13-01761-t001:** Plasma, brain, liver, testis and small intestine pharmacokinetic parameters of repotrectinib 4 h after oral administration of 10 mg/kg repotrectinib to male wild-type, *Abcb1a/1b^−/−^*, *Abcg2^−/−^*, and *Abcb1a/1b;Abcg2^−/−^* mice.

Parameter	Genotype			
	Wild-Type	*Abcb1a/1b^−/−^*	*Abcg2^−/−^*	*Abcb1a/1b;Abcg2^−/−^*
AUC_0–4h_, µg/mL·h	6.99 ± 2.46	9.20 ± 1.64	11.90 ± 1.42 **	14.01 ± 3.18 ***
Fold increase AUC_0–4h_	1.00	1.32	1.70	2.00
C_max_, µg/mL	3.08 ± 1.44	3.63 ± 0.62	5.08 ± 0.49 **	5.23 ± 1.24 **
T_max_, h	0.5 (0.25–1)	0.5 (0.5–1)	0.25 (0.125–0.5)	1 (0.5–2)
C_brain_, ng/g	95.1 ± 23.5	394 ± 102 ***	134 ± 18	2763 ± 928 *** ^(###)^
Fold increase C_brain_	1.00	4.14	1.41	29.1
Brain-to-plasma ratio	0.15 ± 0.03	0.62 ± 0.17 ***	0.14 ± 0.03	2.13 ± 0.21 *** ^(###)^
Fold change ratio	1.00	4.13	0.93	14.2
C_testis_, ng/g	155 ± 55	380 ± 122 ***	166 ± 27	1672 ± 386 *** ^(###)^
Fold increase C_testis_	1.00	2.45	1.07	10.8
Testis-to-plasma ratio	0.24 ± 0.07	0.58 ± 0.14 ***	0.17 ± 0.02	1.35 ± 0.31 *** ^(###)^
Fold change ratio	1.00	2.42	0.71	5.6
C_liver_ µg/g	6.63 ± 1.16	5.58 ± 2.03	6.69 ± 1.21	9.09 ± 2.75
Fold change C_liver_	1.00	0.84	1.01	1.37
Liver-to-plasma ratio	10.5 ± 1.9	8.9 ± 2.4	6.8 ± 0.6 **	7.2 ± 1.2 **
Fold change ratio	1.00	0.85	0.65	0.69
SIC (% of dose)	5.56 ± 1.34	1.13 ± 0.40 ***	4.84 ± 2.21	0.41 ± 0.23 *** ^(##)^
Fold change SIC (% of dose)	1.00	0.20	0.87	0.07
SIC (%)-to-plasma ratio *10^−3^	8.79 ± 2.41	1.86 ± 0.84 ***	4.75 ± 1.54 *	0.30 ± 0.09 *** ^(###)^
Fold change ratio	1.00	0.21	0.54	0.03

Data are presented as mean ± S.D. (n = 6–7). AUC_0–4h_, area under the plasma concentration-time curve; C_max_, maximum concentration in plasma; T_max_, median time point of maximum plasma concentration (range for individual mice); C_brain/liver/SIC_, brain/liver/SIC concentration. SIC (% of dose), drug as percentage of dose present in small intestinal content (SIC), which was expressed as total repotrectinib in SIC divided by total drug administered to mouse. *, *P* < 0.05; **, *P* < 0.01; ***, *P* < 0.001 compared to wild-type mice. ^##^, *P* < 0.01; ^###^, *P* < 0.001 comparing *Abcb1a/1b;Abcg2^−/−^* mice to *Abcb1a/1b^−/−^* mice.

**Table 2 pharmaceutics-13-01761-t002:** Plasma, liver, and small intestine with content (SIWC) pharmacokinetic parameters of repotrectinib in male wild-type and *Oatp1a/1b^−/−^* mice over 2 h after oral administration of 10 mg/kg repotrectinib.

Parameter	Genotype	
	Wild-Type	*Oatp1a/1b^−/−^*
AUC_0–2h_, µg/mL·h	3.92 ± 1.29	8.96 ± 0.77 ***
Fold increase AUC_0–2h_	1.00	2.29
C_max_, µg/mL	2.64 ± 0.99	5.38 ± 0.52 ***
T_max_, h	0.5 (0.25–1)	0.5 (0.5–1)
C_liver_, µg/g	9.84 ± 1.16	15.3 ± 1.6 ***
Fold increase C_liver_	1.00	1.55
Liver-to-plasma ratio	9.16 ± 1.53	4.00 ± 0.54 ***
Fold change ratio	1.00	0.44
C_SIWC_, µg/g	84.74 ± 24.13	121.1 ± 29.2 *
Fold increase C_SIWC_	1.00	1.43
SIWC (% of dose)	17.4 ± 5.6	21.3 ± 3.7
Fold change	1.00	1.22
SIWC (%)-to-plasma ratio·10^−3^	15.8 ± 3.9	5.58 ± 1.09 ***
Fold change ratio	1.00	0.35
SIWC (%)-to-AUC_0–2h_ ratio·10^−3^	4.59 ± 1.04	2.40 ± 0.54 ***
Fold change ratio	1.00	0.52

Data are given as mean ± S.D. (n = 6). AUC_0–2h_, area under the plasma concentration-time curve; C_max_, maximum concentration in plasma; T_max_, median time point of maximum plasma concentration; C_liver_, liver concentration; SIWC, small intestine with content. SIWC (% of dose), drug percentage of dose in small intestine with content, which was expressed as total repotrectinib in SIWC divided by total drug administered to the mouse. *, *P* < 0.05; ***, *P* < 0.001 compared to wild-type mice.

**Table 3 pharmaceutics-13-01761-t003:** Plasma, brain, liver, and small intestine pharmacokinetic parameters of repotrectinib in female wild-type, *Cyp3a^−/−^* and Cyp3aXAV mice over 8 h after oral administration of 10 mg/kg repotrectinib.

Parameter	Genotype		
	Wild-Type	*Cyp3a^−/−^*	Cyp3aXAV
AUC_0–8h_, µg/mL·h	16.28 ± 6.35	37.64 ± 2.83 ***	18.86 ± 3.65 ^###^
Fold change AUC_0–8h_	1.00	2.31	1.16
C_max_, µg/mL	4.80 ± 1.78	8.28 ± 0.93 ***	5.56 ± 0.61 ^##^
T_max_, h	0.5 (0.25–1)	0.5 (0.125–0.5)	0.5 (0.25–0.5)
T_1/2_, h	1.6 ± 0.4	2.5 ± 0.9 *	1.6 ± 0.2 ^#^
C_brain_, ng/g	51.2 ± 17.8	212.8 ± 53.7 ***	80.2 ± 35.3 ^###^
Fold increase C_brain_	1.00	4.16	1.57
Brain-to-plasma ratio	0.26 ± 0.12	0.22 ± 0.08	0.40 ± 0.09
Fold increase ratio	1.00	0.85	1.54
C_liver_, µg/mL	1.22 ± 0.65	5.02 ± 1.92 ***	1.38 ± 0.44 ^###^
Fold increase C_liver_	1.00	4.11	1.13
Liver-to-plasma ratio	5.36 ± 0.54	4.81 ± 0.76	7.10 ± 2.11
Fold increase ratio	1.00	0.90	1.32
SIC (% of dose)	2.08 ± 1.35	5.11 ± 2.83 *	1.68 ± 0.63 ^#^
Fold change	1.00	2.46	0.81

Data are given as mean ± S.D. (n = 6). AUC_0–8h_, area under the plasma concentration-time curve; C_max_, maximum concentration in plasma; T_max_, median time point of maximum plasma concentration; T_1/2_, elimination half-life. C_brain_, brain concentration; SIC, small intestinal content. SIC (% of dose), drug percentage of dose in small intestinal content expressed as total repotrectinib in SIC divided by total drug administered to the mouse. *, *P* < 0.05; ***, *P* < 0.001 compared to wild-type mice; ^#^, *P* < 0.05; ^##^, *P* < 0.01; ^###^, *P* < 0.001 comparing Cyp3aXAV to *Cyp3a^−/−^* mice.

## Data Availability

The data presented in this study are available on request from the corresponding author.

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
