# Peer review of "ABCB1 and ABCG2 Control Brain Accumulation and Intestinal Disposition of the Novel ROS1/TRK/ALK Inhibitor Repotrectinib, While OATP1A/1B, ABCG2, and CYP3A Limit Its Oral Availability"

_pharmaceutics, 2021, doi:10.3390/pharmaceutics13111761_

Round 1
Reviewer 1 Report
The authors have characterized the role of membrane transporters, including the efflux ABC transporters P-gp (Abcb1a/1b) and BCRP (Abcg2) and the uptake SLC transporter OATP (Oatp1a/1b), along with the drug metabolizing enzyme CYP3A (Cyp3a/Cyp3aXAV), in the disposition of orally administered repotrectinib. Based on their in vitro data obtained from cellular systems, in vivo studies with repotrectinib were conducted in knockout mouse models with particular attention on the accumulation of the drug in major organs, including the brain, liver, and small intestine. Apart from a few minor comments listed below, due diligence was achieved by thoroughly analyzing the obtained pharmacokinetic data for a detailed mechanistic understanding. This information was well described in the manuscript for the reader’s understanding. The clinical relevance of this work was also well described, both in the introduction and discussion.
Minor Comments
- In the abstract, for an improved understanding, please provide a quantitative measure describing the extent of repotrectinib transport in vitro.
- Please clarify why repotrectinib transport was only evaluated using an in vitro system overexpressing human ABCB1 but not mouse Abcb1a/1b before advancing to the in vivo mouse models.
- Please be consistent regarding the designation of “r” as efflux ratio, active transport ratio, or relative transport ratio.
- Regarding figure 2, please indicate the number of replicates used to obtain the mean and standard deviation.
- Please explain why either male or female mice were used in certain in vivo studies. For example, female mice were used for the pilot study, whereas male mice were used for the follow-up studies.
- Please comment on the use of DMEM + 10% FBS as the transport assay buffer instead of HBSS. Was protein binding of the drug to FBS a concern?
- Please comment on the use of mean and standard deviation for the pharmacokinetic data (except tmax) in tables 1-3, rather than geometric mean and confidence interval.
- Please include the median (along with the range) for tmax in tables 1-3. Inclusion of this metric would help compare repotrectinib tmax values between mouse strains as depicted in figure 3A and can possibly provide additional mechanistic information.
- Please comment on why the SIC recovery for Abcg2-/- mice was equivalent to the wild-type mice despite a significant increase in AUC and Cmax.
- Please comment on why the role of Abcg2 is minimal (single Abcg2-/- mice) with respect to brain concentration, despite there being a large difference in the magnitude between Abcb1a/1b-/- mice (4.1-fold) and Abcb1a/1b;Abcg2-/- mice (29.1-fold) compared to wild-type mice.
- Please comment on why an increase (rather than an expected decrease) was observed in liver repotrectinib concentration in Oatp1a/1b-/- mice compared to wild-type mice, as shown in figure 4C.
- Please compare the elimination half-life of repotrectinib in Cyp3a-/- mice to that of wild-type mice. Such a comparison may help distinguish the impact of intestinal vs. hepatic CYP3A.
- In the discussion (line 432), please clarify what is meant by “relatively reduced liver distribution of repotrectinib in Abcg2 knockout mice”. The liver concentrations (not liver-to-plasma ratio) in the Abcg2-/- mice (6.69 μg/g) are comparable to those of wild-type mice (6.69 μg/g).
Reviewer 2 Report
The authors analyzed the involvement of the drug efflux carriers ABCB1 and ABCG2 as well as of drug uptake OATP carriers on the transport of the TKI repotrectinib in respective carrier-transfected MDCK cells and respective carrier-deficient knockout mice. In additon the role of CYP3A4 was analyzed in CYP3A4 humanized transgenic mice. The authors found that Abcb1 and Abcg2 restrict the brain penetration of the drug and its intestinal disposition. Oatp1a/1b seem to be invovled in the liver uptake of repotrectinib and CYP3A4 limits the systemic exposure of the drug. The cell and mouse models are well chosen. The paper is nicely written and can be recommended for publication. There are only some minor points that should be additionally discussed.
- The relative role of the endogenous canine ABCB1 of the MDCKII cells and the transfected human ABCB1 carrier on the transcellular transport of repotrectinib is difficult to differenciate. It would help to see the relative species-specific mRNA expression of the respective carrier in this cell model.
- Line 249 should refer to Supplementary Figure 3.
- The 8h pilot experiments shown in Supplementary Figure 1A were done in female mice whereas the subsequent 4h experiments were done in male mice. Although both experiments were done with 10 mg/kg, the absolute plasma concentrations significantly differed. Unfortunately, this difference is neither addressed nor discussed.So please include discussion on this difference.
- The blood-testis-barrier is formed by the Sertoli cells of the seminiferous tubules, while the blood-brain barrier is constituted by the endothelial cells of the brain capillaries. Therefore, these structures are not directly comparable as mentioned in line 278.
- In direct comparison between the wt and the Abcb1a/1b/Abcg2 ko mice the significantly increased AUC in parallel with significantly decreased SIC values does not make sense for me. This point should be more intensively discussed.
- How are Abcb1 and Abcg2 regulated in the Oatp1a/1b ko mice? Up- or downregulation would of course influence the PK data of repotrectinib.
- Looking on the data from the 8h pilot and the 2h study in the Oatp1a/1b ko mice, I am not finally convinced that Oatp1a/1b plays a significant role for the hepatic uptake of the compound. In this case I would expect of course much lower liver concentrations in the knockouts.
Reviewer 3 Report
This article is well-constructed and is very informative for Oncologists. It is valuable to publish in our journal. However, the last sentence of Conclusion is too strong because the in vivo experiments using primary or metastatic brain tumor models are not conducted. The reviewer suggests that the authors should change the expression.
Reviewer 4 Report
This work describes the author studies around Repotrectinib and OATP1A/1B, ABCG2, and CYP3A. The manuscript is very well written and near ready for publication.
An excellent review of the kinome has just been published, this should be added to the introduction - https://www.nature.com/articles/s41573-021-00252-y.
Another key background reference is also missing - https://pubmed.ncbi.nlm.nih.gov/33513356/
The authors could also consider talking about other macrocyclic kinase inhibitors like OD36 and OD38 - https://www.jbc.org/article/S0021-9258(20)37305-1/
Minor edits -
Figure 1 should be made in chemdraw ideally but is acceptable in this form.
line 109 should be 'mL' not 'ml'
line 169 should be 'µL' not 'µl'
line 332 should really be ‘2.5-fold’ not ‘2.45-fold’
line 409 should really be ’14-fold’ not ‘14.2-fold’
Author Response
Please see the attachment.

This manuscript is a resubmission of an earlier submission. The following is a list of the peer review reports and author responses from that submission.